# A New Record for Microbial Perchlorate Tolerance: Fungal Growth in NaClO_4_ Brines and its Implications for Putative Life on Mars

**DOI:** 10.3390/life10050053

**Published:** 2020-04-28

**Authors:** Jacob Heinz, Tim Krahn, Dirk Schulze-Makuch

**Affiliations:** 1Astrobiology Research Group, Center for Astronomy and Astrophysics (ZAA), Technische Universität Berlin, Hardenbergstr. 36, 10623 Berlin, Germany; 2Section Geomicrobiology, German Research Centre for Geosciences (GFZ), 14473 Potsdam, Germany; 3Department of Experimental Limnology, Leibniz-Institute of Freshwater Ecology and Inland Fisheries (IGB), 12587 Stechlin, Germany; 4School of the Environment, Washington State University, Pullman, WA 99163, USA

**Keywords:** Mars, habitability, brines, salts, perchlorate, microorganisms, halotolerant, fungi, yeast, microbial growth

## Abstract

The habitability of Mars is strongly dependent on the availability of liquid water, which is essential for life as we know it. One of the few places where liquid water might be found on Mars is in liquid perchlorate brines that could form via deliquescence. As these concentrated perchlorate salt solutions do not occur on Earth as natural environments, it is necessary to investigate in lab experiments the potential of these brines to serve as a microbial habitat. Here, we report on the sodium perchlorate (NaClO_4_) tolerances for the halotolerant yeast *Debaryomyces hansenii* and the filamentous fungus *Purpureocillium lilacinum*. Microbial growth was determined visually, microscopically and via counting colony forming units (CFU). With the observed growth of *D. hansenii* in liquid growth medium containing 2.4 M NaClO_4_, we found by far the highest microbial perchlorate tolerance reported to date, more than twice as high as the record reported prior (for the bacterium *Planococcus halocryophilus*). It is plausible to assume that putative Martian microbes could adapt to even higher perchlorate concentrations due to their long exposure to these environments occurring naturally on Mars, which also increases the likelihood of microbial life thriving in the Martian brines.

## 1. Introduction

Several observations on Mars, such as characteristic surface morphologies like large fluid-eroded channels, dendritic networks, fluvial valleys, and glacial features [1], and the formation of water-depending minerals like hematite [2], indicate that the planet had a warmer (even though mostly freezing [3]), wetter and more habitable climate in its early history [4]. However, the loss of its magnetic field enabled the solar wind to sputter away large parts of the Martian atmosphere which caused a climate change leading to the dry, cold, and hostile planet that we know today [5,6]. Hence, putative Martian microorganisms would have had to adapt to a gradual decrease in the availability of liquid water. In this process of decreasing water activities, one of the last niches for the occurrence of liquid water would be deposits of hygroscopic salts that absorb water from the thin Martian atmosphere [7]. If these salts can absorb enough water, they will dissolve in the absorbed liquid and form a saturated salt solution (called “brine”) that can be diluted by further water absorption. This process is called deliquescence. Indeed, there is strong evidence for the occurrence of deliquescence processes and of at least temporarily stable brines on Mars [8,9,10].

Several very hygroscopic salts have been detected on Mars that could undergo deliquescence processes [11]. Among these, the most hygroscopic class of salts are perchlorates which have been found in the Martian soil at the Phoenix landing site at concentrations of 0.4–0.6 wt.% [12]. Perchlorate brines have very low freezing points, down to −77.5 °C for a calcium perchlorate solution with an eutectic concentration [13], i.e., the salt concentration causing the most intense freezing point depression. This also enables the provision of liquid water at subzero temperatures and, hence, provides a promising cryo-environment that might serve as a habitat for putative Martian microbes.

On Earth, most of the salt-rich habitats are based on sodium chloride (NaCl), e.g., in the Atacama Desert, Chile, where endolithic cyanobacteria thrive in NaCl crusts by gaining water absorbed by the salt [14,15]. Furthermore, it has also been shown very recently that methanogenic archaea can survive saturated NaCl concentrations and use water exclusively provided by the deliquescence of salt [16]. There are several environments on Earth that also provide high concentrations of salts other than NaCl, e.g., the Dead Sea with increased calcium (0.47 M Ca^2+^) and magnesium (1.98 M Mg^2+^) chloride concentrations (additional to saturated NaCl conditions) [17], the Spotted Lake (Canada) containing high sulfate concentrations (>3 M) [18], the Don Juan Pond (Antarctica) containing 3.7 M CaCl_2_ [19,20], or the Discovery Basin (Mediterranean Sea) containing 5 M MgCl_2_ [21]. While the Dead Sea and the Spotted Lake are inhabited by halophilic microorganisms [18,22], the Don Juan Pond and the Discovery Basin are thought to be sterile environments due to low water activities, high ionic strengths and chaotropic stress [21,23,24].

In contrast to Mars, there are only very few natural environments on Earth where low perchlorate concentrations have been detected, e.g., in the Atacama Desert, Chile [25,26,27], and in the Dry Valleys, Antarctica [28], and no environment is known that consists of concentrated perchlorate brines. Thus, it is appropriate to study these brines in lab experiments to better understand their potential in serving as a habitat for microbial life on Mars. To date, there is only a relatively small body of literature investigating the habitability of brines with perchlorate concentrations above ~ 0.1 M [29,30,31,32,33,34,35,36,37,38]. More studies exist on perchlorate-reducing bacteria (for reviews see [39,40]) and archaea [41] at low perchlorate concentrations below 0.1 M. Prior to this study, the organism found tolerating the highest perchlorate concentration suitable for microbial growth was the halotolerant bacterial strain *Planococcus halocryophilus,* which tolerates 1.1 M sodium perchlorate (NaClO_4_) in its liquid growth medium at 25 °C [30]. Earlier studies reported lower perchlorate tolerances for halotolerant bacteria and halophilic archaea (see Table 1). However, there have been no investigations on the perchlorate tolerance of eukaryotes prior to this study, even though fungi are known to tolerate high concentrations of various other salts [42,43]. 

Here, we report on the NaClO_4_ tolerance of the halotolerant yeast *Debaryomyces hansenii* and the filamentous fungus *Purpureocillium lilacinum*. *D. hansenii* can be found in hyper-saline environments like the Great Salt Lake of Utah or in salterns on the Atlantic coast of Namibia [44]. The yeast has been shown to grow in media containing up to 4 M NaCl [44]. Its high halotolerance results mainly from the accumulation of the compatible solutes glycerol and arabinitol in the exponential growth phase and in the stationary phase, respectively [45]. Additional changes in the metabolism of the yeast during growth under saline conditions are reviewed elsewhere [44,46,47]. In contrast, there is no detailed research on the halotolerance of *P. lilacinum*. Lotlikar and Damare (2018) [48] showed that the fungus can grow in a medium with a salinity of S = 100 (corresponding to a NaCl concentration of 1.9 mol/L), but not at a salinity of S = 250 (5.7 mol/L NaCl). Arpini et al. (2019) [49] found the minimum inhibitory concentration (MIC) for NaCl to be 200 g/L (3.4 mol/L NaCl) for *P. lilacinum*.

## 2. Materials and Methods

### 2.1. Organisms and Culture Conditions

The halotolerant yeast *Debaryomyces hansenii* (DSM 3428) was obtained from the DSMZ (Leibniz Institute DSMZ—German Collection of Microorganisms and Cell Cultures). The yeast cells were grown aerobically at 25 °C (optimum growth temperature) in liquid DMSZ growth medium #90 (3% malt extract, 0.3% soya peptone) with various concentrations of NaClO_4_. The media were prepared by mixing the media components, NaClO_4_ and water, followed by pH adjustment (pH ~5.6) and sterile filtration.

The filamentous fungus *Purpureocillium lilacinum* was found as a contamination at surprisingly high NaClO_4_ concentrations (see Results) during growth experiments with *Planococcus halocryophilus* in liquid DMSZ growth medium #92 (3% tryptic soy broth, 0.3% yeast extract, pH 7.2–7.4, 25 °C) which have been described elsewhere [30]. *P. lilacinum* was isolated and characterized by 18S rDNA sequencing (data not shown). Due to safety restrictions (*P. lilacinum* is categorized as biosafety level S2) no further experiments were conducted with this fungus. 

### 2.2. Determination of Perchlorate Tolerances

The survival and growth of the fungi were determined visibly in 15 mL centrifuge tubes containing the inoculated liquid growth medium, as well as by using light microscopy (Primo Star, Zeiss, equipped with Axio Cam 105 color) and counting colony forming units (CFU) on agar plates (1.5% agar) containing the respective growth medium. The maximum NaClO_4_ concentration suitable for growth was determined through progressive culture adaptation to higher perchlorate concentrations as described previously [30]. In short, 10 µl of a stock culture was used to inoculate 5 ml of liquid growth medium containing 10 wt.% NaClO_4_ (corresponding to 0.9 mol/L). The culture growing from this medium was used to inoculate growth medium containing 15 wt.% NaClO_4_ (1.4 mol/L). This procedure was repeated with increasing NaClO_4_ concentrations in 5 wt.% steps. When no growth could be detected the increments of the NaClO_4_ concentration increase were lowered to 1 wt.%. The highest NaClO_4_ concentration that enabled growth was defined as the perchlorate tolerance of the organism with a technical error of ± 1 wt.% NaClO_4_. All growth experiments were conducted as biological duplicates, i.e., for each NaClO_4_ concentration, two separate samples were inoculated. 

## 3. Results

The perchlorate tolerance of *D. hansenii* was found to be 2.4 M NaClO_4_. Growth curves (until the exponential growth phase) for the samples with this concentration are shown in Figure 1 together with the curves for samples with a concentration of 2.6 M NaClO_4_, in which the cells were dying within 10 days. Additionally, inserted into Figure 1 is a light microscopy image of *D. hansenii* cells after growth in DSMZ medium #90 showing single cells and some small and loosely-bound cell aggregates. Besides a decrease in cell density, no phenotypical changes in the yeast cell morphologies were found when the cells were grown in perchlorate-rich media. This contrasts with earlier studies on the bacterial strain *P. halocryophilus* where cells grown in perchlorate-rich media formed large cell clusters [30]. 

*Purpureocillium lilacinum* was not investigated in detail due to safety restrictions (see Section 2.1). However, it was found as a contaminant growing in liquid growth medium having NaClO_4_ concentrations of up to 1.9 M, which we interpret to be its uppermost tolerance to NaClO_4_. 

## 4. Discussion

Before the detection of significant amounts of perchlorate on Mars in 2008 [12], the scientific interest in determining the habitability of concentrated perchlorate brines was low due to their practical non-existence in natural habitats on Earth. Since 2008, the number of studies investigating this question has increased but is still insufficient for understanding the potential for life in Martian perchlorate-containing brines. Table 1 (see Introduction) lists the two record holders for each domain of life regarding their NaClO_4_ tolerance as described in the literature thus far. 

The study presented here is the first one describing significant perchlorate tolerances for eukaryotes. Several other studies described growth of non-fungal organisms in perchlorate solutions with concentrations lower than the ones listed in Table 1 [31,33,34,35,36,37,38]. For example, Oren et al. (2014) [31] found that several halophilic archaea of the family Halobacteriaceae (*Halobacterium* strain NRC-1, *Hbt. salinarum* R1, *Haloferax volcanii*, *Hfx. mediterranei*, *Hfx. denitrificans*, *Hfx. gibbonsii*, *Haloarcula marismortui*, and *Har. vallismortis*) and the bacterium *Halomonas elongata* grew well in NaCl-based media supplemented with a perchlorate concentration of up to 0.4 M. However, among these, only *Hfx. mediterranei* was also able to grow in 0.6 M NaClO_4_. Al Soudi et al. (2017) [33] found that the halotolerant bacterial strains *Marinococcus halophilus*, *Halomonas venusta*, and *Bacillus licheniformis* grew robustly at 0.5 M NaClO_4_, while only *H. venusta* also showed substantial growth at 1.0 M NaClO_4_. Furthermore, it has been shown that methanogenic archaea can metabolize and produce methane at perchlorate concentrations of up to 0.4 mol/L [38].

Therefore, the two fungi investigated in this study, *D. hansenii* and *P. lilacinum*, have by far the highest tolerances to NaClO_4_ among all microorganisms investigated to date. The tolerance for *D. hansenii* (2.4 M NaClO_4_) is more than twice as high as for the bacterial strain *P. halocryophilus* that has been holding the NaClO_4_ tolerance record (1.1 M NaClO_4_) prior to this study [30]. As the two fungi described in this study are the first ones ever investigated regarding their perchlorate tolerance, it is plausible to assume that other fungi (e.g., the extremely halotolerant black yeast *Hortaea werneckii*, or the obligately halophilic *Wallemia ichthyophaga* [43]) might tolerate even higher perchlorate concentrations, which we plan to investigate in upcoming experiments. 

The perchlorate tolerance data available in the literature to date (Table 1) convey the impression that fungal species have a more efficient perchlorate defense machinery than bacteria and archaea. However, more research on microbial and fungal perchlorate tolerances and the adaptation mechanisms applied by the fungi grown in perchlorate-rich growth media is needed to confirm or discard this hypothesis. Furthermore, the effect of additional stress factors typical for Mars on the fungal survival and growth in perchlorate brines should prompt further investigations. The most relevant stressors in this context are low temperatures and pressures, high radiation levels, and stress induced by the higher chaotropicity and ionic strengths of ions from other perchlorate salts like magnesium or calcium perchlorate, which probably represent the majority of perchlorate salts on Mars [50,51].

Since there are no natural perchlorate-rich environments existing on Earth and thought not to have existed in the past, there is no obvious adaptation mechanism for the two fungal species investigated in this study to have adapted to these high NaClO_4_ concentrations. If life on Mars exists in perchlorate brines, we may speculate that these microorganisms might have evolved tolerance to much higher perchlorate concentrations, as they—in contrast to Earth—would have been under natural selection pressures on Mars to achieve higher perchlorate tolerances. We recommend that this hypothesis should be tested in future Mars missions via life detection experiments in saline and perchlorate-rich environments, such as locations where the presumably deliquescence-driven “recurring slope lineae” (RSL) [10] occur.

## Figures and Tables

**Figure 1 life-10-00053-f001:**
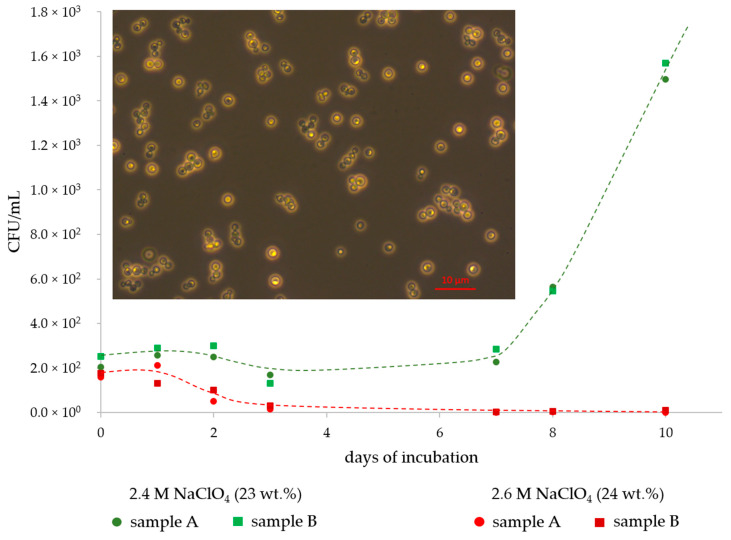
Growth curves of *D. hansenii* at 25 °C in liquid growth media with a sodium perchlorate (NaClO_4_) concentration of 2.4 M (green line and symbols) and 2.6 M (red line and symbols). Experiments were run in biological duplicates (samples A and B). Microbial growth occurred only in the 2.4 M NaClO_4_ samples while the cells in the 2.6 M NaClO_4_ samples died within 10 days. Cells of *D. hansenii* after growth in liquid growth medium are shown in the implemented image.

**Table 1 life-10-00053-t001:** Sodium perchlorate (NaClO_4_) tolerances (in mol/L, wt.% [w/w], and wt./vol.% [w/v]) for the two organisms of each domain of life tolerating the highest perchlorate concentrations reported to date.

Domain	Organism	NaClO_4_ Tolerance			Literature
		(mol/L)	(wt.%)	(wt./vol.%)	
Archaea	*Haloferax mediterranei*	0.6	6.8	7.3	[31]
	*Halorubrum lacusprofundi*	0.8	8.9	9.8	[32]
Bacteria	*Halomonas venusta*	1.0	10.9	12.2	[33]
	*Planococcus halocryophilus*	1.1	12.0	13.6	[30]
Eukarya (Fungi)	*Purpureocillium lilacinum*	1.9	19.0	23.5	This study
	*Debaryomyces hansenii*	2.4	23.0	29.9	This study

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
