# Peer review of "A New Record for Microbial Perchlorate Tolerance: Fungal Growth in NaClO4 Brines and its Implications for Putative Life on Mars"

_life, 2020, doi:10.3390/life10050053_

Round 1

Reviewer 1 Report

The title is somewhat misleading. It may suggest that both brines and fungal do exist on Mars.  What about “A New Record for Microbial Perchlorate Tolerance: application to Martian Brines”.

Liebensteiner et al. found that the archeon, Archaeoglobus fulgidus, can also reduce perchlorate, another living being potentially adapted to the Martian environment. The reference should be added.

Liebensteiner et al. (2013). Archaeal (Per)Chlorate Reduction at High Temperature: An Interplay of Biotic and Abiotic Reactions. Science 340, 85-87; DOI: 10.1126/science.1233957.

Author Response

The title is somewhat misleading. It may suggest that both brines and fungal do exist on Mars.  What about “A New Record for Microbial Perchlorate Tolerance: application to Martian Brines”.

We thank the reviewer for pointing out the issue with the title. We reformulated to a better version: “A New Record for Microbial Perchlorate Tolerance: Fungal Growth in NaClO4 Brines and its Implications for Putative Life on Mars”

Liebensteiner et al. found that the archeon, Archaeoglobus fulgidus, can also reduce perchlorate, another living being potentially adapted to the Martian environment. The reference should be added.

Liebensteiner et al. (2013). Archaeal (Per)Chlorate Reduction at High Temperature: An Interplay of Biotic and Abiotic Reactions. Science 340, 85-87; DOI: 10.1126/science.1233957.

The reference was incorporated into the revised manuscript in lines 71-72.

We thank the reviewer for his/her helpful comments and edits. We marked all changes in the manuscript referring to his/her review in yellow.

Reviewer 2 Report

My commendations on a well put together paper.

I only have a couple of issues:

The authors repeatedly use the term 'halotolerance' in reference to their experiments, yet only test resistance to one salt. Several other papers on the area of salt effects have started using 'halo-' to only refer to sodium chloride (halide) salts, and use other terms for other salts, so this could cause confusion. I suspect this is the most efficient shorthand, but I don't think it's quite accurate. Perhaps the authors could think of something more accurate - I don't know if 'perchlorotolerance' rolls of the tongue - if not then perhaps a short explanatory sentence early in the paper to explain this.

Lines 168-170 - I'm pretty sure there's fairly extensive literature on the fact that fungi are generally more halotolerant than other microbes, the authors could mention this and include some references.

Lines 182-187 - I find these final questions somewhat flabby and arbitary, and don't believe it would be possible to test this in terrestrial laboratories. Given the evolutionary history of fungi, I doubt we can test whether significant amounts of organic matter are the key part of their halotolerance (although it does tend to be the most energetically favourable food source, perhaps allowing additional metabolic energy for halo-resistive strategies), or if that is just serendipitous adaptation. The first part ("That begs the question...") is far too specific, and the later part far too general. I think the authors should reword.

Author Response

My commendations on a well put together paper.

I only have a couple of issues:

The authors repeatedly use the term 'halotolerance' in reference to their experiments, yet only test resistance to one salt. Several other papers on the area of salt effects have started using 'halo-' to only refer to sodium chloride (halide) salts, and use other terms for other salts, so this could cause confusion. I suspect this is the most efficient shorthand, but I don't think it's quite accurate. Perhaps the authors could think of something more accurate - I don't know if 'perchlorotolerance' rolls of the tongue - if not then perhaps a short explanatory sentence early in the paper to explain this.

Thank you for this clarification. We kept the term “halotolerance” in cases where it refers to literature describing microbial NaCl tolerances. Wherever we describe our results we changed the term to “perchlorate tolerance”, or sometimes more specifically to “NaClO4 tolerance”.

Lines 168-170 - I'm pretty sure there's fairly extensive literature on the fact that fungi are generally more halotolerant than other microbes, the authors could mention this and include some references.

To the best of our knowledge, halophilic archaea are more known for their high halotolerances (and here again referring mainly to NaCl tolerances) than fungi. For, example ref. [41] notes “Wallemia ichthyophaga is presently the most halophilic fungus known to date. It shows preference for media supplemented with salt over media with high concentrations of glucose and it grows only between 10 % by mass per volume of NaCl and saturation (32 % NaCl), equivalent to aw of 0.77. Such obligate halophily is common in archaea, but unique in fungi.” The here described fungus, Wallemia ichthyophaga, is mentioned in the manuscript in line 172. It might be true that some fungi have a wider halotolerance range (e.g. Hortaea werneckii growing in salt-free media as well as close to NaCl saturation) when not being obligatory halophilic, but this is also true for many halotolerant bacteria (e.g. for Staphylococcus arlettae HPSSN35C isolated from saline soil growing between 0 and 6 M NaCl [52]). We are not aware of any study describing that fungi can generally tolerate higher salt (mainly NaCl) concentrations than other microbes (bacteria and archaea). If the reviewer has specific references in mind proofing the fact that fungi generally have higher halotolerances, we would be happy to incorporate them in the manuscript.

[41] Gunde-Cimerman, N.; Zalar, P. Extremely halotolerant and halophilic fungi inhabit brine in solar salterns around the globe. Food Technol. Biotech. 2014, 52, 170–179.

[52] Nanjani, S.G.; Soni, H.P. Characterization of an extremely halotolerant Staphylococcus arlettae HPSSN35C isolated from Dwarka Beach, India. J. Basic Microbiol. 2014, 54, 843–850, doi:10.1002/jobm.201200690.

Lines 182-187 - I find these final questions somewhat flabby and arbitary, and don't believe it would be possible to test this in terrestrial laboratories. Given the evolutionary history of fungi, I doubt we can test whether significant amounts of organic matter are the key part of their halotolerance (although it does tend to be the most energetically favourable food source, perhaps allowing additional metabolic energy for halo-resistive strategies), or if that is just serendipitous adaptation. The first part ("That begs the question...") is far too specific, and the later part far too general. I think the authors should reword.

We deleted the final question of the discussion and reworded the last paragraph.

We thank the reviewer very much for his/her very helpful comments and edits. We marked all changes in the manuscript referring to his/her review in green.
